# Deep Structured Prediction for Facial Landmark Detection

**Lisha Chen**[1]**, Hui Su**[1,2]**, Qiang Ji**[1]
[1]Rensselaer Polytechnic Institute, [2]IBM Research
chenl21@rpi.edu, huisuibmres@us.ibm.com, jiq@rpi.edu

## Abstract

Existing deep learning based facial landmark detection methods have achieved excellent performance. These methods, however, do not explicitly embed the structural dependencies among landmark points. They hence cannot preserve the geometric relationships between landmark points or generalize well to challenging conditions or unseen data. This paper proposes a method for deep structured facial landmark detection based on combining a deep Convolutional Network with a Conditional Random Field. We demonstrate its superior performance to existing state-of-the-art techniques in facial landmark detection, especially a better generalization ability on challenging datasets that include large pose and occlusion.

## 1 Introduction

Facial landmark detection is to automatically localize the fiducial facial landmark points around facial components and facial contour. It is essential for various facial analysis tasks such as facial expression analysis, headpose estimation and face recognition. With the development of deep learning techniques, traditional facial landmark detection approaches that rely on hand-crafted low-level features have been outperformed by deep feature based approaches. The purely deep learning based methods, however, cannot effectively capture the structural dependencies among landmark points. They hence cannot perform well under challenging conditions, such as large head pose, occlusion, and large expression variation. Probabilistic graphical models such as Conditional Random Fields (CRFs), have been widely applied to various computer vision tasks. They can systematically capture the structural relationships among random variables and perform structured prediction. Recently, there have been works that combine deep models with CRF to simultaneously leverage convolutional neural networks' (CNNs) representation power and CRF's structure modeling power [10, 9, 51]. Their combination has yielded significant performance improvement over methods that use either CNN or CRF alone. These works so far are mainly applied to classification tasks such as semantic image segmentation. Besides classification, some works apply the CNN and CRF model to human pose [41, 12, 11] and facial landmark detection [2, 44] . To simplify computational complexity, the CRF models are typically of special structure (e.g. tree structure), moreover, they employ approximate learning and inference criteria. In this work, we propose to combine CNN with a fully-connected CRF to jointly perform facial landmark detection in regression framework.

Compared to the existing works, the contributions of our work are summarized as follows:
1) We introduce the fully-connected CNN-CRF that produces structured probabilistic prediction of facial landmark locations.
2) Our model explicitly captures the structure relationship variations caused by pose and deformation, unlike some previous works that combine CNN with CRF using a fixed pairwise relationship.
3) We use an alternating method and derive closed-form solutions in the alternating steps for learning and inference, unlike previous works that use approximate methods such as energy minimization which ignores the partition function for learning and mean-field for inference. And instead of using discriminative criterion or other approximate loss functions, we employ negative log likelihood (NLL)

loss function, without any assumption.

4) Experiments on benchmark face alignment datasets demonstrate the advantages of the proposed method in achieving better prediction accuracy and generalization to challenging or unseen data than current state-of-the-art (SoA) models.

## 2    Related Work

### 2.1    Facial Landmark Detection

Classic facial landmark detection methods including Active Shape Model (ASM) [14, 28], Active Appearance Model (AAM) [13, 24, 27, 36], Constrained Local Model (CLM) [25, 37], and Cascade Regression [8, 6, 53, 7, 46] rely on hand-crafted shallow image features and are usually sensitive to initializations. They are outperformed by modern deep learning based methods.

Using deep learning for face alignment was first proposed in [39] and achieved better performance than classic methods. This purely deep appearance based approach uses a deep cascade convolutional network and coordinate regression in each cascade level. Later on, more work using *purely deep appearance based* framework for coordinate regression has been explored. Tasks-constrained deep convolutional network (TCDCN) [50] was proposed to jointly optimize facial landmark detection with correlated tasks such as head pose estimation and facial attribute inference. Mnemonic Descent Method (MDM) [42], an end-to-end trainable deep convolutional Recurrent Neural Network (RNN), was proposed where the cascade regression was implemented by RNN. Recently, heatmap learning based methods established new state-of-the-art for face alignment and body pose estimation [41, 30, 43]. And most of these face alignment methods [5, 44] follow the architecture of Stacked Hourglass [30]. The stacked modules refine the network predictions after each stack. Different from direct coordinate regression, it predicts a heatmap with the same size as the input image. *Hybrid deep methods* combine deep models with face shape models. One strategy is to directly predict 3D deformable parameters instead of landmark locations in a cascaded deep regression framework, e.g. 3D Dense Face Alignment (3DDFA) [54] and Pose-Invariant Face Alignment (PIFA) [23]. Another strategy is to use the deformable model as a constraint to limit the face shape search space thus to refine the predictions from the appearance features, e.g. Convolutional Experts Constrained Local Model (CE-CLM) [48].

### 2.2    Structured Deep Models

To produce structured predictions, some works combine deep models with graphical models. Early works like [31] jointly train a CNN and a graphical model for image segmentation. Do et al.[16] introduced NeuralCRF for sequence labeling. And various works are explored for other tasks. For instance, Jain et al. [22] and Eigen et al. [18]'s work for image restoration, Yao et al. and Morin et al.'s work [47, 29] for language understanding, Yoshua et al., Peng et al. and Jaderberg et al.'s work [3, 32, 21] for handwriting or text recognition. Recently, for human body pose estimation, Chen et al.[10] use CNN to output image dependent part presence as the unary term and spatial relationship as the pairwise potential in a tree-structured CRF and uses Dynamic Programming for inference. Tompson et al. [41, 40] jointly trained a CNN and a fully-connected MRF by using the convolution kernel to capture pairwise relationships among different body joints and an iterative convolution process to implement the belief propagation. The idea of using convolution to implement message passing has also been explored in [12], where structure relationships at the body joint feature level rather than the output level are captured in a bi-directional tree structured model. And the work of Chu et al.[12] is applied to face alignment [44] to pass messages between facial part boundary feature maps. As an extension to [12], [11] models structures in both output and hidden feature layers in CNN. Similarly, for image segmentation, DeepLab [9] uses fully connected CRF with binary cliques and mean-field inference, and [26] uses efficient piecewise training to avoid repeated inference during training. In [51], the CRF mean-field inference is implemented by RNN and the network is end-to-end trainable by directly optimizing the performance of the mean-field inference. Using RNN to implement message passing has also been applied to facial action unit recognition [15]. In [20], the MRF deformable part model is implemented as a layer in a CNN.

**Comparison.**   Compared to previous models serving similar purposes such as [12, 11, 44] that assume a tree structured model with belief propagation as inference method, we use a fully-connected

model. With a fully connected model, we don't need to specify a certain tree structured model, letting the model learn the strong or weak relationships from data, thus this method is more generalizable to different tasks. And the works [41, 12, 11, 44, 51] use convolution to implement the pairwise term and the message passing process. The pairwise term, once trained, is independent of the input image, thus cannot capture the pairwise constraint variations across different conditions like target object rotation and object shape. However, we explicitly capture the object pose, deformation variations. Moreover, they employ approximate methods such as energy minimization ignoring the partition function for learning and mean-field for inference. In this paper we do exact learning and inference, capturing the full covariance of the joint distribution of facial landmarks given deformable parameters. Lastly, compared to the traditional CRF models [33, 34], the weights for each unary terms in our model are also outputs of the neural network whose inverse quantifies heteroscedastic aleatoric uncertainty of the unary prediction.

# 3 Method

This section presents the proposed structured deep probabilistic facial landmark detection model. In this model, the joint probability distribution of facial landmark locations and deformable parameters are captured by a conditional random field model.

## 3.1 Model definition

Denote the face image as $\mathbf{x}$, the 2D facial landmark locations as $\mathbf{y}$, each landmark is $\mathbf{y}_i, i = 1, \ldots, N$. The deformable model parameters that capture pose, identity and expression variation are denoted as $\zeta$. The model parameter we want to learn is denoted as $\Theta$. Assuming $\zeta$ is marginally dependent on $\mathbf{x}$ but conditionally independent of $\mathbf{x}$ given $\mathbf{y}$, the graphical model is shown in Fig. 1.

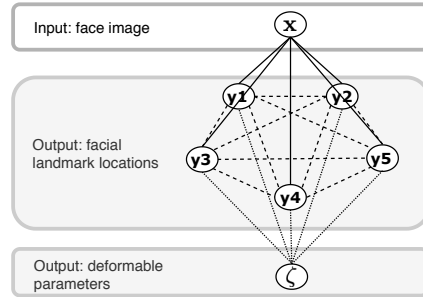

Based on this definition and assumption, the joint distribution of landmarks $\mathbf{y}$ and deformable parameters $\zeta$ conditioned on the face image $\mathbf{x}$ can be formulated in a CRF framework and written as

Figure 1: The graphical model. Dashed, dotted, solid lines represent dependencies between pairs of landmarks, landmark and deformable parameters, landmarks and face image, respectively.

$$p_\Theta(\mathbf{y}, \zeta \mid \mathbf{x}) = \frac{1}{Z_\Theta(\mathbf{x})} \exp\{ -\sum_{i=1}^N \phi_{\theta_1}(\mathbf{y}_i \mid \mathbf{x}) - \sum_{i=1}^N \sum_{j=i+1}^N \psi_{C_{ij}}(\mathbf{y}_i, \mathbf{y}_j, \zeta) \} \tag{1}$$

where $\Theta = [\theta_1, C_{ij}]$, $\theta_1$ is neural network parameter, $C_{ij}$ is a $2 \times 2$ symmetric positive definite matrix that captures the spatial relationships between a pair of landmark points, $\mathbf{y}_i$ and $\mathbf{y}_j$. $Z_\Theta(\mathbf{x})$ is the partition function. $\phi_{\theta_1}(\mathbf{y}_i \mid \mathbf{x})$ is the unary energy function with parameter $\theta_1$ and $\psi_{C_{ij}}(\mathbf{y}_i, \mathbf{y}_j, \zeta)$ is the triple-wise energy function with parameter $C_{ij}$.

## 3.2 Energy functions

We define the unary and triple-wise energy in Eq.(2) and Eq.(3) respectively.

$$\phi_{\theta_1}(\mathbf{y}_i \mid \mathbf{x}) = \frac{1}{2}[\mathbf{y}_i - \mu_i(\mathbf{x}, \theta_1)]^T \Sigma_i^{-1}(\mathbf{x}, \theta_1)[\mathbf{y}_i - \mu_i(\mathbf{x}, \theta_1)] \tag{2}$$

$$\psi_{C_{ij}}(\mathbf{y}_i, \mathbf{y}_j, \zeta) = [\mathbf{y}_i - \mathbf{y}_j - \mu_{ij}(\zeta)]^T C_{ij}[\mathbf{y}_i - \mathbf{y}_j - \mu_{ij}(\zeta)] \tag{3}$$

where $\mu_i(\mathbf{x}, \theta_1)$ and $\Sigma_i(\mathbf{x}, \theta_1)$ are the outputs of the CNN that represent mean and covariance matrix of each landmark given the image $\mathbf{x}$. $\mu_{ij}(\zeta)$ represents the expected difference between two landmark locations. It is fully determined by the 3D deformable face shape parameters $\zeta$, which contains rigid parameters: rotation $R$ and scale $S$, and non-rigid parameters $\mathbf{q}$. $\begin{bmatrix} \mu_{ij}(\zeta) \\ 1 \end{bmatrix} =$

$\frac{1}{\lambda}SR(\bar{\mathbf{y}}_i^{3d} + \Phi_i\mathbf{q} - \bar{\mathbf{y}}_j^{3d} - \Phi_j\mathbf{q})$, where $\bar{\mathbf{y}}^{3d}$ is the 3D mean face shape, $\Phi$ is the bases of deformable model, they are learned from data. The deformable parameters $\zeta = [S, R, \mathbf{q}]$ are jointly estimated with 2D landmark locations during inference. In this work, we assume weak perspective projection model. $S$ is a $3 \times 3$ diagonal matrix that contains 2 independent parameters $s_x, s_y$ as scaling factor (encode the camera intrinsic parameters) for column and row respectively. While $R$ is a $3 \times 3$ orthonormal matrix with 3 independent parameters $\gamma_1, \gamma_2, \gamma_3$ as the pitch, yaw, roll rotation angle. Note that the translation vector is canceled by taking the difference of two landmark points.

### 3.3 Learning and Inference

We propose to implement the conditional probability distribution in Eq. (1) with a CNN-CRF model. As shown in Fig. 2, the CNN with parameter $\theta_1$ outputs mean $\mu_i(\mathbf{x}, \theta_1)$ and covariance matrix $\Sigma_i(\mathbf{x}, \theta_1)$ for each facial landmark $\mathbf{y}_i$, which together forms the unary energy function $\phi_{\theta_1}(\mathbf{y}_i \mid \mathbf{x})$. A fully-connected (FC) graph with parameter $C_{ij} \succ 0$ gives the triple-wise energy $\psi_{C_{ij}}(\mathbf{y}_i, \mathbf{y}_j, \zeta)$, if given $\zeta$ as well as the output from the unary, the FC can output $\mathrm{E}(\mathbf{x}, \zeta, \Theta)$ and $\Lambda_p(\mathbf{x}, \zeta, \Theta)$, the mean and precision matrix for the conditional distribution $p_\Theta(\mathbf{y} \mid \zeta, \mathbf{x})$. The FC can be implemented as another layer following the CNN. Combining the unary and the triple-wise energy, we obtain the joint distribution $p_\Theta(\mathbf{y}, \zeta \mid \mathbf{x})$. However, direct inference of $\mathbf{y}^*, \zeta^*$ from $p_\Theta(\mathbf{y}, \zeta \mid \mathbf{x})$ is difficult, therefore we iteratively infer from conditional distributions $p_\Theta(\mathbf{y} \mid \zeta, \mathbf{x})$ and $p_\Theta(\zeta \mid \mathbf{y})$.

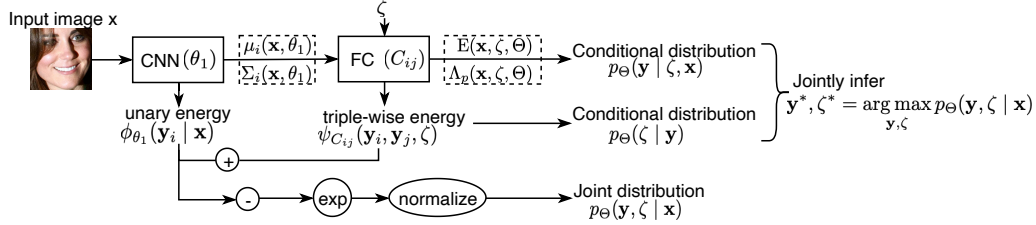

Figure 2: Overall flowchart of the proposed CNN-CRF model.

### Mean and Precision matrix

During learning and inference, we need to compute conditional probability $p_\Theta(\mathbf{y} \mid \zeta, \mathbf{x})$. By using the quadratic unary and triple-wise energy function, the distribution $p_\Theta(\mathbf{y} \mid \zeta, \mathbf{x})$ is a multivariate Gaussian distribution that can be written as

$$p_\Theta(\mathbf{y} \mid \zeta, \mathbf{x}) = \frac{1}{Z'_\Theta(\mathbf{x})} \exp\{-\sum_{i=1}^{N}\phi_{\theta_1}(\mathbf{y}_i \mid \mathbf{x}) - \sum_{i=1}^{N}\sum_{j=i+1}^{N}\psi_{C_{ij}}(\mathbf{y}_i, \mathbf{y}_j, \zeta)\}$$

$$= \exp\{\frac{1}{2}\ln|\Lambda_p(\mathbf{x}, \Theta, \zeta)| - \frac{1}{2}[\mathbf{y} - \mathrm{E}(\mathbf{x}, \Theta, \zeta)]^T\Lambda_p(\mathbf{x}, \Theta, \zeta)[\mathbf{y} - \mathrm{E}(\mathbf{x}, \Theta, \zeta)]\}$$

(4)

where $Z'_\Theta(\mathbf{x})$ is the partition function. $\mathrm{E}(\mathbf{x}, \Theta, \zeta)$ and $\Lambda_p(\mathbf{x}, \Theta, \zeta)$ is the mean and precision matrix of the multivariate Gaussian distribution. They are computed exactly during learning and inference. The mean $E$ can be computed by solving the linear system of equations $\Lambda_p E = b$ where $\Lambda_p$, the precision matrix, is a symmetric positive definite matrix that can be directly computed from the coefficient in the unary and pairwise term as shown in Eq. (5), and $b$ can be computed from Eq. (5).

$$\Lambda_p = \begin{bmatrix} \Lambda_{p11} & \dots & \Lambda_{p1N} \\ \vdots & \ddots & \vdots \\ \Lambda_{pN1} & \dots & \Lambda_{pNN} \end{bmatrix}, \begin{cases} \Lambda_{pii} = \Sigma_i^{-1} + \sum_{j \neq i} C_{ij} \\ \Lambda_{pij} = -C_{ij} \end{cases} \quad b = \begin{bmatrix} b_1 \\ b_2 \\ \vdots \\ b_N \end{bmatrix}, b_i = \Sigma_i^{-1}\mu_i + \sum_{j \neq i} C_{ij}\mu_{ij} \quad (5)$$

From Eq.(5) we can see that the final inference result $E_i$ is a combination of $\mu_i$ and $\mu_j + \mu_{ij}, j \in \{1, \dots, N\}, j \neq i$. To solve this linear system of equations, we use direct method for exact solution with a fast implementation by Cholesky factorization that requires $\mathcal{O}(N^3)$ FLOPs. For a practical implementation of the determinant to avoid numerical issues, we again use the Cholesky factorization of $\Lambda_p$ to get $LL^T = \Lambda_p$, then we compute the log determinant by $\ln|\Lambda_p| = 2\sum \ln diag(L)$ where $diag(\cdot)$ takes the diagonal element of a matrix.

## Learning

During learning, our goal is to optimize $\Theta$ given training data $\mathcal{D} = \{\mathbf{x}_m, \mathbf{y}_m, m = 1, \ldots, M\}$. We directly optimize the inference performance. Note that we don't have ground truth label for $\zeta$, where $\zeta = \{\zeta_1, \ldots, \zeta_m\}$. We use an alternating method, based on the current $\Theta^t$, $\hat{\mathbf{y}}^t = \mathrm{E}(\mathbf{x}, \Theta^t, \zeta^t)$, optimize $\zeta$ by

$$\zeta_m^{t+1} = \arg\min_{\zeta_m} -\ln p_{\Theta^t}(\hat{\mathbf{y}}_m^t, \zeta_m \mid \mathbf{x}_m) = \arg\min_{\zeta_m} \psi_{C_{ij}^t}(\hat{\mathbf{y}}_{mi}^t, \hat{\mathbf{y}}_{mj}^t, \zeta_m) \tag{6}$$

Then based on current $\zeta^t$, optimize $\Theta$ by

$$\Theta^{t+1} = \arg\min_{\Theta} Loss = \arg\min_{\Theta} -\sum_{m=1}^{M} \ln p_{\Theta}(\mathbf{y}_m, \zeta_m^t \mid \mathbf{x}_m) = \arg\min_{\Theta} -\sum_{m=1}^{M} \ln p_{\Theta}(\mathbf{y}_m \mid \zeta_m^t, \mathbf{x}_m)$$

$$= \arg\min_{\Theta} \sum_{m=1}^{M} -\frac{1}{2}\ln|\Lambda_p(\mathbf{x}_m, \Theta, \zeta_m^t)| + \frac{1}{2}[\mathbf{y}_m - \mathrm{E}(\mathbf{x}_m, \Theta, \zeta_m^t)]^T \Lambda_p(\mathbf{x}_m, \Theta, \zeta_m^t)[\mathbf{y}_m - \mathrm{E}(\mathbf{x}_m, \Theta, \zeta_m^t)] \tag{7}$$

The algorithm for this problem is designed to first set $C_{ij} = \mathbf{0}$ and optimize $\theta_1$, the CNN parameter. Then set $C_{ij} = 0.01I$ and optimize $\zeta$, then fix a subset of parameters from $\Theta$ and optimize the others alternately, whose pseudo code is shown in Algorithm 1.

---

**Algorithm 1:** Learning CNN-CRF

---

**Input:** training data $\{\mathbf{x}_m, \mathbf{y}_m, m = 1, \ldots, M\}$;
**Initialization:** parameters $\Theta^0 = \{\theta_1^0 = randn, C_{ij}^0 = \mathbf{0}\}, t = 0$ ;
**while** *not converge* **do**
$\quad\mid\quad \theta_1^{t+1} = \theta_1^t - \eta_1^t \frac{\partial Loss}{\partial \theta_1}$ ; $t = t + 1$;
**end**
$\hat{\mathbf{y}}_m^t = \mathrm{E}(\mathbf{x}_m, \Theta^t, \zeta^t), C_{ij}^t = 0.01I$;
**while** *not converge* **do**
$\quad$ **Stage 1:** Fix parameters $\Theta = \Theta^t$, optimize $\zeta$ by Eq. (6);       $\triangleright$ Optimize deformable parameters
$\quad$ **while** *not converge* **do**
$\quad\quad\mid\quad \zeta_m^{t+1} = \arg\min_{\zeta_m} \psi_{C_{ij}^t}(\hat{\mathbf{y}}_{mi}^t, \hat{\mathbf{y}}_{mj}^t, \zeta_m), \hat{\mathbf{y}}^{t+1} = \mathrm{E}(\mathbf{x}, \Theta, \zeta^{t+1}), t = t + 1$;
$\quad$ **end**
$\quad \Theta^t = \Theta$
$\quad$ **Stage 2:** Fix $\zeta = \zeta^t, C_{ij} = C_{ij}^t$, update $\theta_1$ using Eq. (7);                $\triangleright$ Update CNN parameters
$\quad\quad$ **while** *not converge* **do**
$\quad\quad\mid\quad \theta_1^{t+1} = \theta_1^t - \eta_1^t \frac{\partial Loss}{\partial \theta_1}$ ; $t = t + 1$;
$\quad$ **end**
$\quad [\zeta^t, C_{ij}^t] = [\zeta, C_{ij}]$
$\quad$ **Stage 3:** Fix $\zeta = \zeta^t, \theta_1 = \theta_1^t$, update $C_{ij}$ using Eq. (7);                $\triangleright$ Update CRF parameters
$\quad\quad$ **while** *not converge* **do**
$\quad\quad\mid\quad C_{ij}^{t+1} = C_{ij}^t - \eta_2^t \frac{\partial Loss}{\partial C_{ij}}$; $t = t + 1$;
$\quad$ **end**
$\quad [\zeta^t, \theta_1^t] = [\zeta, \theta_1]$
**end**

---

## Inference

The inference problem is a joint inference of $\zeta, \mathbf{y}$ for each input face image $\mathbf{x}$, defined in Eq. (8)

$$\mathbf{y}^*, \zeta^* = \arg\max_{\mathbf{y}, \zeta} \ln p_{\Theta}(\mathbf{y}, \zeta \mid \mathbf{x}) \tag{8}$$

We use an alternating method. Based on current $\mathbf{y}^t$, optimize $\zeta^t$ by (see supplementary):

$$\zeta^t = \arg\max_{\zeta} \ln p_{\Theta}(\mathbf{y}^t, \zeta \mid \mathbf{x}) = \arg\min_{\zeta} \sum_{i=1}^{N} \sum_{j=i+1}^{N} \psi_{C_{ij}}(\mathbf{y}_i^t, \mathbf{y}_j^t, \zeta) \tag{9}$$

Then based on current $\zeta^t$, optimize $\mathbf{y}^{t+1}$ by:

$$\mathbf{y}^{t+1} = \arg\max_{\mathbf{y}} \ln p_{\Theta}(\mathbf{y}, \zeta^t \mid \mathbf{x}) = \arg\max_{\mathbf{y}} \ln p_{\Theta}(\mathbf{y} \mid \zeta^t, \mathbf{x}) = \mathrm{E}(\mathbf{x}, \Theta, \zeta^t) \tag{10}$$

The inference algorithm is shown in Algorithm 2.

---

**Algorithm 2:** Inference for CNN-CRF

---
**Input:** face image $\mathbf{x}$
**Initialization:** $\mathbf{y}_i^0 = \mu_i, i = 1, \ldots, N$ , $t = 0$;
**while** *not converge* **do**
$\quad$ Update $\zeta$ by Eq. (9). $\zeta^t = \arg\min_\zeta \sum_{i=1}^N \sum_{j=i+1}^N \psi_{C_{ij}}(\mathbf{y}_i^t, \mathbf{y}_j^t, \zeta)$;
$\quad$ Update $\mathbf{y}$ by Eq. (10). $\mathbf{y}^{t+1} = \mathrm{E}(\mathbf{x}, \Theta, \zeta^t)$;
$\quad$ $t = t + 1$ ;
**end**

---

## 4    Experiments

**Datasets.** We evaluate our methods on popular benchmark facial landmark detection datasets, including 300W [35], Menpo [49], COFW [6], 300VW [1].

*300W* has 68 landmark annotation. It contains 3837 faces for training and 300 indoor and 300 outdoor faces for testing.

*Menpo* contains images from AFLW and FDDB with landmark re-annotation following the 68 landmark annotation scheme. It has two subsets, Menpo-frontal which has 68 landmark annotations for near frontal faces (6679 samples) and Menpo-profile which has 39 landmark annotations for profile faces (2300 samples). We use it as a test set for cross dataset evaluation.

*COFW* has 1345 training samples and 507 testing samples, whose facial images are all partially occluded. The original dataset is annotated with 29 landmarks. We use the COFW-68 test set [19] which has 68 landmarks re-annotation for cross dataset evaluation.

*300VW* is a facial video dataset with 68 landmarks annotation. It contains 3 scenarios: 1) constrained laboratory and naturalistic well-lit conditions; 2) unconstrained real-world conditions with different illuminations, dark rooms, overexposed shots, etc.; 3) completely unconstrained arbitrary conditions including various illumination, occlusions, make-up, expression, head pose, etc. We use the test set for cross dataset evaluation.

**Evaluation metrics.** We evaluate our algorithm using the standard normalized mean error (NME) and the Cumulative Errors Distribution (CED) curve. Besides, the area-under-the-curve (AUC) and the failure rate (FR) for a maximum error of 0.07 are reported. Same as in [5], the NME is defined as the average point-to-point Euclidean distance between the ground truth ($\mathbf{y}_{gt}$) and predicted ($\mathbf{y}_{pred}$) landmark locations normalized by the ground truth bounding box size $d = \sqrt{w_{bbox} * h_{bbox}}$, $\mathrm{NME} = \frac{1}{N} \sum_{i=1}^N \frac{||\mathbf{y}_{pred}^{(i)} - \mathbf{y}_{gt}^{(i)}||_2}{d}$. Based on the NME in the test dataset, we can draw a CED Curve with NME as the horizontal axis and percentage of test images as the vertical axis. Then the AUC is computed as the area under that curve for each test dataset.

**Implementation details.** To make a fair comparison with the SoA purely deep learning based methods [5], we use the same training and testing procedure for 2D landmark detection. The 3D deformable model was trained on the 300W-train dataset or 300W-LP dataset by structure from motion [4]. For CNN, we use 4 stacks of Hourglass with the same structure as [5], each stack followed by a softmax layer to output a probability map for each facial landmark. From the probability map, we compute mean $\mu_i$ and covariance $\Sigma_i$. And we use additional softmax cross entropy loss and L1 loss on the mean [38] to assist training which shows better performance empirically.

**Training procedure:** The initial learning rate $\eta_1$ is $10^{-4}$ for 15 epochs using a minibatch of 10, then dropped to $10^{-5}$ and $10^{-6}$ after every 15 epochs and keep training until convergence. The learning rate $\eta_2$ is set to $10^{-3}$. We applied random augmentations such as random cropping, rotation, etc. We first train the method on 300W-LP [54] dataset which is augmented from the original 300W dataset for large yaw pose. And then we fine-tune on the original 300W train dataset.

**Testing procedure:** We follow the same testing procedure as [5]. The face is cropped using the ground truth bounding box defined in 300W. The cropped face is rescaled to $256 \times 256$ before passed to the network. For the Menpo-profile dataset, the annotation scheme is different, we use the overlapping 26 points for evaluation, i.e., removing points other than the 2 endpoints on the face contour and the eyebrow respectively and removing the 5th point on the nose contour.

### 4.1    Comparison with existing approaches

In Table 1, we compare with some most recent best results reported, in the 300W protocol that trains on LFPW-train, HELEN-train, AFW and tests on LFPW-test, HELEN-test, ibug and use NME normalized with inter-ocular/pupil distance as the metric.

In Table 2, we compare with other baseline facial landmark detection algorithms, including purely deep learning based methods such as TCDCN [50] and FAN [5] as well as hybrid methods such as CLNF [2] and CE-CLM [48]. The results for these methods are evaluated using the code provided by the authors in the same experiment protocol, i.e., same bounding box and same evaluation metrics. The CED curves on the 300W testset are shown in Fig. 3a.

Table 1: Comparison with SoA methods on 300W dataset using 300W protocol (NME normalized with inter-ocular/pupil distance %)

| Subset / Method | Com. | Chal. | Full |
|---|---|---|---|
| Inter-ocular distance | | | |
| MDM [42] | - | - | 4.05 |
| RDR [45] | 5.03 | 8.95 | 5.80 |
| SAN [17] | 3.34 | 6.60 | 3.98 |
| LAB (4-stack) [44] | 2.98 | 5.19 | 3.49 |
| Our method (4-stack) | **2.93** | **4.84** | **3.30** |
| Inter-pupil distance | | | |
| MDM [42] | 4.83 | 10.14 | 5.88 |
| LAB (4-stack) [44] | 4.20 | 7.41 | 4.92 |
| Our method (4-stack) | **4.06** | **6.98** | **4.63** |

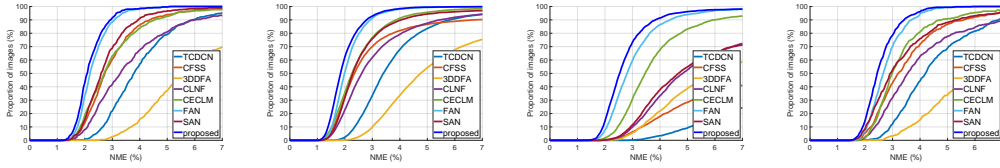

(a) 300W testset     (b) Menpo-frontal dataset     (c) Menpo-profile dataset     (d) COFW-68 testset
Figure 3: CED curves on different datasets (better viewed in color and magnified)

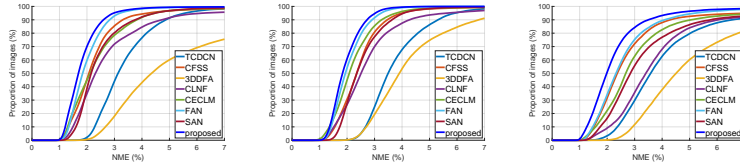

(a) 300VW category1     (b) 300VW category2     (c) 300VW category3
Figure 4: CED curves on 300VW testset (better viewed in color and magnified)

**Cross-dataset Evaluation**
Besides 300W testset, we evaluate the proposed method on Menpo dataset, COFW-68 testset, 300VW testset for cross dataset evaluation. The results are shown in Table 2 for Menpo and COFW-68 dataset and Table 3 for 300VW dataset. And the CED curves are shown in Fig. 3b, 3c, 3d respectively. The method is trained on 300W-LP and fine-tuned on 300W Challenge train set for 68 landmarks. We can see that compared to the results on 300W testset and Menpo-frontal dataset, where the SoA methods attaining saturating performance as mentioned in [5], for cross-dataset evaluation in more challenging conditions such as COFW with heavy occlusion and Menpo-profile with large pose, the proposed method shows better generalization ability with a significant performance improvement. On the other hand, the proposed method shows smallest failure rate (FR) on all evaluated datasets.

Table 2: Within and cross dataset prediction results (%)

| Dataset / Method | 300W-test | | | Menpo-frontal | | | Menpo-profile | | | COFW-68 test | | |
|---|---|---|---|---|---|---|---|---|---|---|---|---|
| Metric | NME | AUC | FR | NME | AUC | FR | NME | AUC | FR | NME | AUC | FR |
| TCDCN [50] | 4.15 | 42.1 | 4.83 | 4.04 | 46.2 | 5.84 | 13.96 | 5.9 | 75.61 | 4.71 | 35.8 | 8.68 |
| CFSS [52] | 3.09 | 56.7 | 1.83 | 3.91 | 57.4 | 9.75 | 15.04 | 15.2 | 58.87 | 3.79 | 49.0 | 4.34 |
| 3DDFA [54] | 6.90 | 20.6 | 30.00 | 6.57 | 28.7 | 24.57 | 8.37 | 20.5 | 41.43 | 8.13 | 18.2 | 43.79 |
| CLNF [2] | 4.22 | 47.6 | 6.67 | 3.74 | 55.4 | 5.82 | 8.32 | 27.8 | 27.65 | 4.75 | 42.9 | 10.65 |
| CE-CLM [48] | 3.05 | 56.9 | 2.33 | 2.78 | 63.3 | 1.66 | 4.63 | 45.2 | 7.17 | 3.36 | 52.4 | 2.37 |
| FAN (reported in [5]) | - | 66.9 | - | - | 67.5 | - | - | - | - | - | - | - |
| SAN [17] | 2.86 | 59.7 | 1.00 | 2.95 | 61.9 | 3.11 | 8.80 | 29.0 | 28.65 | 3.50 | 51.9 | 3.94 |
| our method | 2.21 | 68.1 | 0.17 | 2.01 | 71.0 | 0.16 | 3.03 | 60.0 | 1.96 | 2.55 | 63.2 | 0.00 |

## 4.2 Analysis

In this section, we report the results of sensitivity analysis and ablation study. If not specified, analysis is performed on test datasets with models trained on 300W-LP and fine-tuned on 300W train set.

Table 3: 300VW testset prediction results for cross-dataset evaluation (%)

| Dataset | 300VW-category1 | | | 300VW-category2 | | | 300VW-category3 | | |
|---|---|---|---|---|---|---|---|---|---|
| Metric<br>Method | NME | AUC | FR | NME | AUC | FR | NME | AUC | FR |
| TCDCN [50] | 3.49 | 51.2 | 1.74 | 3.80 | 45.8 | 1.76 | 4.45 | 43.8 | 8.85 |
| CFSS [52] | 2.44 | 67.0 | 1.66 | 2.49 | 64.3 | 0.77 | 3.26 | 60.5 | 5.18 |
| 3DDFA [54] | 5.80 | 32.4 | 24.50 | 4.44 | 39.2 | 8.82 | 5.48 | 31.6 | 18.26 |
| CLNF [2] | 3.34 | 60.4 | 4.31 | 2.98 | 60.0 | 3.02 | 4.73 | 47.1 | 7.74 |
| CE-CLM [48] | 2.54 | 65.7 | 1.58 | 2.39 | 66.0 | 0.61 | 3.61 | 56.4 | 5.69 |
| FAN (reported in [5]) | - | 72.1 | - | - | 71.2 | - | - | 64.1 | - |
| SAN [17] | 2.58 | 64.5 | 1.10 | 2.57 | 63.2 | 0.42 | 4.06 | 52.9 | 7.19 |
| our method | 1.91 | 73.3 | 0.36 | 1.97 | 71.6 | 0.04 | 2.50 | 67.4 | 1.68 |

**Sensitivity to challenging conditions.** We evaluate different methods on challenging conditions caused by either high noise, low resolution, or different initializations in Fig. 5. Generally, the proposed CNN-CRF model is more robust under challenging conditions compared to a pure CNN model with the same structure, i.e. the CNN-CRF model with $C_{ij} = \mathbf{0}$.

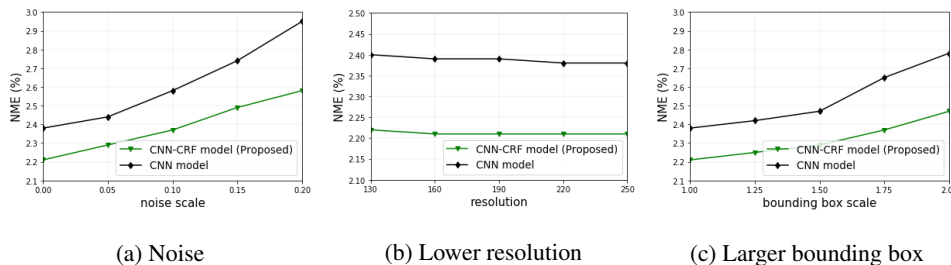

(a) Noise  (b) Lower resolution  (c) Larger bounding box

Figure 5: Prediction error sensitivity to challenging conditions

**Ablation Study.** The improvement of the proposed method lies in two aspects. On the one hand, the proposed softmax + L1 mean loss + Gaussian negative log likelihood (NLL) loss gives better results empirically. On the other hand, the joint training of the CNN-CRF model with the assistance of the deformable model captures structured relationships with pose and deformation awareness. To analyze the effect of the proposed method, in Table 4, we evaluate the performance of a plain CNN prediction, the 3D deformable model fitting to the ground truth, and the joint CNN-CRF prediction accuracy.

Table 4: Ablation study on 300W testset (%)

| Method | NME | AUC | FR |
|---|---|---|---|
| Plain CNN with softmax cross entropy loss | 2.38 | 65.9 | 0.50 |
| Plain CNN with softmax + L1 mean loss + Gaussian NLL loss (proposed loss) | 2.30 | 67.4 | 0.50 |
| Separately trained CNN and CRF with proposed loss | 2.23 | 67.8 | 0.50 |
| Deformable model fitting | 1.39 | 79.8 | 0.00 |
| Jointly trained CNN-CRF with proposed loss (proposed method) | 2.21 | 68.1 | 0.17 |

## 5   Conclusion

In this paper, we propose a method combining CNN with a fully-connected CRF model for facial landmark detection. Compared to the state-of-the-art purely deep learning based methods, our method explicitly captures the structured relationships between different facial landmark locations. Compared to previous methods that combine CNN with CRF for human body pose estimation that learn a fixed pairwise relationship representation for different test samples implemented by convolution, our methods capture the structure relationship variations caused by pose and deformation. Moreover, we use a fully-connected model instead of a tree-structured model, obtaining a better representation ability. Lastly, compared to previous methods that do approximate learning such as omitting the partition function and inference such as mean-field method, we perform exact learning and inference, thus able to provide a better structured uncertainty. Experiments on benchmark datasets demonstrate that the proposed method outperforms the existing state-of-the-art methods, in particular under challenging conditions, for both within dataset and cross dataset.

**Acknowledgment** The work described in this paper is supported in part by NSF award IIS #1539012 and by RPI-IBM Cognitive Immersive Systems Laboratory (CISL), a center in IBM's AI Horizon Network.

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
