[Supplementary Material · supplement-nips19-v5.pdf]

# Deep Structured Prediction for Facial Landmark Detection (Supplementary Material)

**Lisha Chen**[1], **Hui Su**[1,2], **Qiang Ji**[1]
[1]Rensselaer Polytechnic Institute, [2]IBM Research
chenl21@rpi.edu, huisuibmres@us.ibm.com, jiq@rpi.edu

## A Details of implementation

### A.1 Evaluation

Our results for other methods are obtained using the code and default trained models provided by the authors. Specifically, the references and link to the official code that we use to evaluate are provided in Table 1.

Table 1: References and link to the code for compared algorithms

| Method | link to the official code |
|---|---|
| TCDCN [12] | https://github.com/zhzhanp/TCDCN-face-alignment [11] |
| CFSS [15] | https://github.com/zhusz/CVPR15-CFSS [14] |
| 3DDFA [16] | http://www.cbsr.ia.ac.cn/users/xiangyuzhu/projects/3DDFA/main.htm [17] |
| CLNF [1] | https://github.com/TadasBaltrusaitis/OpenFace [2] |
| CE-CLM [10] | https://github.com/TadasBaltrusaitis/OpenFace [2] |
| FAN [5] | https://github.com/1adrianb/2D-and-3D-face-alignment [4] |
| SAN [7] | https://github.com/D-X-Y/landmark-detection/tree/master/SAN [6] |

### A.2 Implementation details

We use Tensorflow 1.14 as our deep learning framework, and one NVIDIA GeForce RTX 2080 Ti GPU for training. The optimizer we use is Adam. For evaluation on all datasets except for Menpo-profile dataset, we train on 300W-LP and fine-tuned on 300W trainset and the 3D deformable model was trained on 300W-trainset. For evaluation on Menpo-profile dataset, we train on 300W-LP and the 3D deformable model was trained on 300W-LP dataset. Our code will be available at https://github.com/lisha-chen/Deep-structured-facial-landmark-detection.

## B Additional Experiment Results

### B.1 Prediction visualization

We show in Fig. 1 the predictions of different methods on some benchmark datasets and their ground truth annotations for a visual comparison. On Fig. 1b, only predictions of visible landmark points on the profile faces as well as the ground truth annotations are drawn for a clearer comparison. We can see from the results that hybrid methods such as CFSS and CE-CLM preserves plausible face shape but their predictions may be wrong for large pose. While purely deep learning based method such as FAN can handle large poses for some samples or points but does not preserve the face shape.

(a) COFW68-test dataset         (b) Menpo-profile dataset

Figure 1: Challenging example images with red dots and lines representing ground truth annotations and white dots and lines representing predictions from different methods.

## C Derivation

### C.1 Mean-field approximation

In this section, we show some derivations to analyze the effect of some approximate learning and inference methods, thus to compare with our exact learning and inference to show the pros and cons. We focus on the part of estimating $\Theta^{t+1}$ given a known $\zeta^t$ as shown in Eq. (1)

$$
\begin{aligned}
\Theta^{t+1} &= \arg\min_{\Theta} -\sum_{m=1}^{M} \ln p_\Theta(\mathbf{y}_m, \zeta_m^t \mid \mathbf{x}_m) = \arg\min_{\Theta} -\sum_{m=1}^{M} \ln p_\Theta(\mathbf{y}_m \mid \zeta_m^t, \mathbf{x}_m) \\
&= \arg\min_{\Theta} \sum_{m=1}^{M} -\frac{1}{2}\ln|\Lambda_p(\mathbf{x}_m, \Theta, \zeta_m^t)| + \frac{1}{2}[\mathbf{y}_m - \mathrm{E}(\mathbf{x}_m, \Theta, \zeta_m^t)]^T \Lambda_p(\mathbf{x}_m, \Theta, \zeta_m^t)[\mathbf{y}_m - \mathrm{E}(\mathbf{x}_m, \Theta, \zeta_m^t)]
\end{aligned}
$$
(1)

In Eq. (1), we estimate $\Theta^{t+1}$ by minimizing the negative log joint probability of facial landmarks $p_\Theta(\mathbf{y}_m \mid \zeta_m^t, \mathbf{x}_m)$. However, in some works [9], it is proposed to use mean-field approximation to speed up the learning and inference process especially for large scale problems such as image labeling.

In our work, we can also use mean-field inference. We thus derive the result from the mean-field inference and compare with the result from the exact method.

#### C.1.1 General equation for mean-field inference

To derive the general equation for mean-field inference. Denote the distribution we want to approximate as $P(y)$, $P(y) = \frac{1}{Z}\tilde{P}(y)$ and the mean-field distribution as $Q(y)$ where $y$ is the random variable. By minimizing the KL divergence between $Q$ and $P$, we have Eq. (2) which is also given in [3].

$$
Q_i(y_i) = \frac{1}{Z_i}\exp\{E_{Q(y/y_i)}[\ln \tilde{P}(y)]\}
$$
(2)

Because

$$D_{KL}(Q||P) = E_Q[\ln \frac{Q}{P}]$$

$$= -E_Q[\ln \tilde{P}(y)] + E_Q[\ln Z] + \sum_i E_{Q_i}[\ln Q_i(y_i)]$$

$$= -E_Q[\ln \tilde{P}(y)] + \ln Z + \sum_i E_{Q_i}[\ln Q_i(y_i)] \tag{3}$$

$$s.t. \int Q_i(y_i)dy_i = 1$$

where the first term

$$-E_Q[\ln \tilde{P}(y)] = -\int \prod_i Q_i(y_i)[\ln \tilde{P}(y)]dy$$

$$= -\int Q_i(y_i) \int \prod_{j \neq i} Q(y_j)[\ln \tilde{P}(y)]dy_{j \neq i}dy_i \tag{4}$$

$$= -\int Q_i(y_i)E_{Q(y/y_i)}[\ln \tilde{P}(y)]dy_i$$

In order to minimize $D_{KL}(Q||P)$, we need to take derivative w.r.t. each $Q_i$ using the Lagrangian method and set it to zero for a constrained optimization problem.

$$\mathcal{L} = D_{KL}(Q||P) + \lambda \sum_i (\int Q_i(y_i)dy_i - 1) \tag{5}$$

Therefore,

$$\frac{\partial \mathcal{L}}{\partial Q_i(y_i)} = -E_{Q(y/y_i)}[\ln \tilde{P}(y)] + \ln Q_i(y_i) + 1 + \lambda = 0 \tag{6}$$

$$Q_i(y_i) = \exp(-\lambda - 1)\exp\{E_{Q(y/y_i)}[\ln \tilde{P}(y)]\} = \frac{1}{Z_i}\exp\{E_{Q(y/y_i)}[\ln \tilde{P}(y)]\} \tag{7}$$

### C.1.2  Mean-field approximation of multivariate Gaussian distribution

In our case, where $P$ is a multivariate Gaussian distribution, therefore $Q$ is also a multivariate Gaussian distribution by minimizing the KL divergence. Therefore

$$E_q[y_i] = \arg\max_{y_i} Q_i(y_i) = \arg\max_{y_i} E_{Q(y/y_i)}[\ln \tilde{P}(y)] \tag{8}$$

Thus to compute $E_q[y_i]$, we take derivative of $E_{Q(y/y_i)}[\ln \tilde{P}(y)]$ w.r.t. $y_i$ and set it to zero. Then we have

$$\frac{\partial E_{Q(y/y_i)}[\ln \tilde{P}(y)]}{\partial y_i} = E_{Q(y/y_i)}\Big[\frac{\partial \ln \tilde{P}(y)}{\partial y_i}\Big] = 0 \tag{9}$$

$$E_{Q(y/y_i)}\Big[\Lambda_{pi}(y - E_p[y])\Big] = 0 \ for \ i = 1,\dots,N. \tag{10}$$

$$E_{Q(y/y_i)}\Big[\sum_j \Lambda_{pij}(y_j - E_p[y_j])\Big] = 0 \ for \ i = 1,\dots,N. \tag{11}$$

$$\sum_j \Lambda_{pij}(E_q[y_j] - E_p[y_j]) = \Lambda_{pi}(E_q[y] - E_p[y]) = 0 \ for \ i = 1,\dots,N. \tag{12}$$

From Eq. (12), we can get a linear system of equations to solve for $E_q[y]$, that is

$$\Lambda_p(E_q[y] - E_p[y]) = \mathbf{0} \tag{13}$$

The solution to Eq. (13) is $E_q[y] = E_p[y]$. Now we compute the precision matrix of the mean-field distribution $\Lambda_q$. From Eq. (7) we have

$$Q_i(y_i) = \frac{1}{Z_i}\exp\{E_{Q(y/y_i)}[\ln \tilde{P}(y)]\} = \frac{1}{Z_i}\exp\{E_{Q(y/y_i)}[-\frac{1}{2}(y - E_p[y])^T\Lambda_p(y - E_p[y])]\} \tag{14}$$

Since $E_q = E_p$, $E_{Q_j(y_j)}[\Lambda_{pij}(y_j - E_p[y_j])] = 0$,

$$Q_i(y_i) = \frac{1}{Z_i} \exp\{-\frac{1}{2}(y_i - E_p[y_i])^T \Lambda_{pii}(y_i - E_p[y_i])\} \tag{15}$$

Therefore $E_q = E_p$, $Cov_q[y_i] = \Lambda_{pii}^{-1}$. This conclusion is also given in [3] that the model variable mean is correct but the variance of $Q$ is determined by the direction of smallest variance of $P$.

And the strength of mean-field is that it is a simpler form without computing the inverse or determinant for the whole matrix $\Lambda_p$. And during learning, if we use the result of the mean-field to directly optimize the mean-field prediction performance as did in [13], it will affect the gradient for $C_{ij}, \mu_i, \Sigma_i$. During inference, the predicted mean is the same as the exact method but the predicted covariance is different, it will be a diagonal (or band) matrix. This is known as a major failing of mean-field method that it underestimates the uncertainty of model variables and does not capture their covariance (structured uncertainty) [8].

## C.2 Optimize deformable model parameters given landmarks

**To solve** $\zeta = [\tilde{S}, R, \mathbf{q}]$, where $R = \begin{bmatrix} r_1 \\ r_2 \\ r_3 \end{bmatrix}$, $\tilde{S} = \frac{1}{\lambda}\begin{bmatrix} s_1 & 0 & 0 \\ 0 & s_2 & 0 \\ 0 & 0 & 1 \end{bmatrix}$. We first solve $R$ and $\tilde{S}$, then $\mathbf{q}$ and iterate this process. Let $M^{2\times 3} = \frac{1}{\lambda}\begin{bmatrix} s_1 & 0 \\ 0 & s_2 \end{bmatrix}\begin{bmatrix} r_1 \\ r_2 \end{bmatrix}$. Rearrange the problem in matrix format

$$M = \arg\min_M \sum_{i=1}^{N}\sum_{j=i+1}^{N}[\mathbf{y}_{ij} - M\bar{\mathbf{y}}_{ij}^{3d}]^T C_{ij}[\mathbf{y}_{ij} - M\bar{\mathbf{y}}_{ij}^{3d}]$$
$$= \arg\min_M [\mathbf{y}_{IJ} - append(M\bar{\mathbf{y}}_{IJ}^{3d})]^T \Lambda_{C_{IJ}}[\mathbf{y}_{IJ} - append(M\bar{\mathbf{y}}_{IJ}^{3d})] \tag{16}$$

where $\mathbf{y}_{IJ} = \begin{bmatrix} \mathbf{y}_{12} \\ \mathbf{y}_{13} \\ \vdots \\ \mathbf{y}_{N-1,N} \end{bmatrix}$, $append(M\bar{\mathbf{y}}_{IJ}^{3d}) = \begin{bmatrix} M\bar{\mathbf{y}}_{12}^{3d} \\ M\bar{\mathbf{y}}_{13}^{3d} \\ \vdots \\ M\bar{\mathbf{y}}_{N-1,N}^{3d} \end{bmatrix}$. $\bar{\mathbf{y}}^{3d}$ is known, therefore $append(M\bar{\mathbf{y}}_{IJ}^{3d})$ could be written as a linear function of $M$, denoted as $f(M)$. We rearrange $M = \begin{bmatrix} m_1 \\ m_2 \end{bmatrix}$ to be a $6 \times 1$ vector $M = \begin{bmatrix} m_1^T \\ m_2^T \end{bmatrix}$. Then $f(M) = F^{[N(N-1)]\times 6}M^{6\times 1}$.

where

$$F = \begin{bmatrix} \bar{\mathbf{y}}_{IJ1}^{3dT} & \mathbf{0} \\ \mathbf{0} & \bar{\mathbf{y}}_{IJ1}^{3dT} \\ \bar{\mathbf{y}}_{IJ2}^{3dT} & \mathbf{0} \\ \mathbf{0} & \bar{\mathbf{y}}_{IJ2}^{3dT} \\ \vdots & \vdots \\ \bar{\mathbf{y}}_{IJ\frac{N(N-1)}{2}}^{3dT} & \mathbf{0} \\ \mathbf{0} & \bar{\mathbf{y}}_{IJ\frac{N(N-1)}{2}}^{3dT} \end{bmatrix}, \Lambda_{C_{IJ}} = \begin{bmatrix} C_{IJ1} & \mathbf{0} & \cdots & \mathbf{0} \\ \mathbf{0} & C_{IJ2} & \vdots & \vdots \\ \vdots & \cdots & \ddots & \mathbf{0} \\ \mathbf{0} & \cdots & \mathbf{0} & C_{IJ\frac{N(N-1)}{2}} \end{bmatrix} \tag{17}$$

Let $L = [\mathbf{y}_{IJ}^{2d} - append(M\bar{\mathbf{y}}_{IJ}^{3d})]^T \Lambda_{C_{IJ}}[\mathbf{y}_{IJ}^{2d} - append(M\bar{\mathbf{y}}_{IJ}^{3d})]$ and take the derivative and set to zero we have

$$\frac{\partial L}{\partial M} = \frac{\partial(\mathbf{y}_{IJ}^{2d} - f(M))}{\partial M}\frac{\partial L}{\partial(\mathbf{y}_{IJ}^{2d} - f(M))}$$
$$= F^T \Lambda_{C_{IJ}}(\mathbf{y}_{IJ}^{2d} - FM) = 0 \tag{18}$$

Therefore $F^T \Lambda_{C_{IJ}}FM = F^T \Lambda_{C_{IJ}}\mathbf{y}_{IJ}^{2d}$, $M = (F^T \Lambda_{C_{IJ}}F)^{-1}F^T \Lambda_{C_{IJ}}\mathbf{y}_{IJ}^{2d}$. Given $M$, $R$ is obtained via SVD.

Note that: $F^T \Lambda_{C_{IJ}} F$ is invertible when $\Lambda_{C_{IJ}} \succ 0$. Because if we take $\mathbf{v} \in \mathbb{R}^6, \mathbf{v} \neq \mathbf{0}$, then $F\mathbf{v} \neq \mathbf{0}$ since if $F\mathbf{v} = \mathbf{0}, \mathbf{v}^T = [\mathbf{v}_1^T, \mathbf{v}_2^T]$ then there exists a plane with normal $\mathbf{v}_1$ or $\mathbf{v}_2$ such that the 3D landmarks $\bar{\mathbf{y}}_i^{3d}, i = 2, \ldots, N$ are on the same plane, which is not true in our case. $\mathbf{v}^T F^T \Lambda_{C_{IJ}} F\mathbf{v} > 0$, therefore $F^T \Lambda_{C_{IJ}} F \succ 0$.

**To solve q:** Given $R, S, \lambda$, we can solve $\mathbf{q}$ linearly as below

$$
\begin{aligned}
\mathbf{q} &= \arg\min_{\mathbf{q}} \sum_{i=1}^{N} \sum_{j=i+1}^{N} [\mathbf{y}_{ij} - \mu_{ij}(\zeta)]^T C_{ij} [\mathbf{y}_{ij} - \mu_{ij}(\zeta)] \\
&= \arg\min_{\mathbf{q}} \sum_{i=1}^{N} \sum_{j=i+1}^{N} [\mathbf{y}_{ij} - M(\bar{\mathbf{y}}_{ij}^{3d} + \Phi_{ij}\mathbf{q})]^T C_{ij} [\mathbf{y}_{ij} - M(\bar{\mathbf{y}}_{ij}^{3d} + \Phi_{ij}\mathbf{q})] \\
&= \arg\min_{\mathbf{q}} [\mathbf{y}_{IJ} - append(M(\bar{\mathbf{y}}_{IJ}^{3d} + \Phi_{IJ}\mathbf{q}))]^T \Lambda_{C_{IJ}} [\mathbf{y}_{IJ} - append(M(\bar{\mathbf{y}}_{IJ}^{3d} + \Phi_{IJ}\mathbf{q}))]
\end{aligned} \tag{19}
$$

take the derivative and set to zero we have

$$
\begin{aligned}
\frac{\partial L}{\partial \mathbf{q}} &= \frac{\partial (\mathbf{y}^{2d} - g(\mathbf{q}))}{\partial \mathbf{q}} \frac{\partial L}{\partial (\mathbf{y}^{2d} - g(\mathbf{q}))} \\
&= G^T \Lambda_{C_{IJ}} (\mathbf{y}_{IJ}^{2d} - g(\mathbf{q})) = 0
\end{aligned} \tag{20}
$$

$g(\mathbf{q}) = G\mathbf{q} + T, G = append(M\Phi_{IJ}), T = append(M\bar{\mathbf{y}}_{IJ}^{3d})$. Therefore $G^T \Lambda_{C_{IJ}} G\mathbf{q} = G^T \Lambda_{C_{IJ}} (\mathbf{y}_{IJ}^{2d} - T), \mathbf{q} = (G^T \Lambda_{C_{IJ}} G)^{-1} G^T \Lambda_{C_{IJ}} (\mathbf{y}_{IJ}^{2d} - T)$. We conduct the two steps of solving $M$ and $\mathbf{q}$ iteratively until convergence.

Note that: $G^T \Lambda_{C_{IJ}} G$ is invertible when $\Lambda_{C_{IJ}} \succ 0$.