[Reviews · NeurIPS 2019]

Reviewer 1



The integration of convnets with the conditional random fields to model the structural dependencies of facial landmarks during face alignment is nice contribution. Previously proposed methods in this direction were hybrid systems (eg. OpenFace versions) and not fully integrated. The authors evaluate on multiple datasets (300W, 300W-Video, Menpo & COFW-68) and compare results with other methods. Both inter- and cross-dataset performance are provided. Writing is good, easy to follow. Couple of concerns, mainly about the evaluation protocol: 1) The authors compare to methods with available code, and they re-ran the codes on the benchmark datasets. The reported results (in Table 1) in some cases are lower than the ones reported in the original papers (see results in FAN[5]). This leaves the reader wondering on what is the source of this discrepancy? Were different parameters used during the reproduction? This need more discussion and would be important to state the original performance scores. 2) The evaluation metric was taken from [5], which is a modification of the original 300W benchmark metric. The original metric from 300W dataset normalizes landmark errors to the inter-ocular distance and widely accepted and used in the computational face community. The study of [5] modified the normalization the the bounding box size, and was adopted only in a few papers. The two normalization is not compatible and the scores are not comparable to the methods that use the original protocol. See for example (Dong et al, 2018) for a list of recent SotA methods. Without reporting scores using the original metric is difficult to judge how the proposed technique compares to the state-of-the-art. Also, among the compared methods there is only one recent study from 2017, the others are 4-5 years old techniques. Dong, Xuanyi, et al. "Style aggregated network for facial landmark detection." Proceedings of the IEEE Conference on Computer Vision and Pattern Recognition. 2018. 3) It's not clear where is the 3D model (that was used in the study) coming from. Eq. 4 refers to a linear 3D subspace (\Phi) that provides the the rigid (S,R) and non-rigid (q) parameters. Was this subspace learned from a different 3D dataset or inferred from 2D using structure from motion?

Reviewer 2



Originality: The paper proposes a new method that combines a fully-connected CRF which is completely learned and inferred with a CNN for structured predictions of facial landmarks. Its novelty is incremental as variants of CRFs have been widely used and combined with CNNs previously, including for prediction of landmarks, albeit using different formulations than the one proposed. Quality: The authors show improvements over several existing methods and compare on 4 datasets, where their method performs the best. However, they omit comparisons against many of the latest state of the art methods. I would like to see how their method compares against these latest methods. Clarity: The paper is well written and clear. Significance: The paper is of incremental value in a niche application area.

Reviewer 3



The paper presents an approach for facial landmark detection/alignment based on a combination of a Convolutional Neural Network and a Conditional Random Field, to impose structure on predicted landmarks. Strengths: 1. The work is novel 2. The paper is well presented and clear 3. Results are very promising Weaknesses: 1. Unclear experimental methodology. The paper states that 300W-LP is used to train the model, but later it is claimed same procedure is used as was used for baselines. Most baselines do not use 300W-LP dataset in their training. Is 300W-LP used in all experiments or just some? If it is used in all this would provide an unfair advantage to the proposed method. 2. Missing link to similar work on Continuous Conditional Random Fields [Ristovski 2013] and Continuous Conditional Neural Fields [Baltrusaitis 2014] that has a similar structure of the CRF and ability to perform exact inference. 3. What is Gaussian NLL? This seems to come out of nowhere and is not mentioned anywhere in the paper, besides the ablation study? Trivia: Consider replacing "difference mean" with "expected difference" between two landmarks (I believe it would be clearer)

[Author Response · NeurIPS 2019]

Common issues: As per reviewers request, we compare with most recent SoA methods using official code with the same metric. Results are shown in Table 1. We extend our work (CNN part) to use 4 stacks of HourGlass modules with intermediate supervision, this gives improved performance as shown in Table 1, which clearly shows our method outperforms the best compared state-of-the-art (SoA) methods on most datasets given the similar number of parameters as [5]. And our method works particularly well on challenging datasets such as Menpo-profile, COFW-68 and 300VW-category3, 300W-train challenge set, significantly outperforming the compared SoA methods.

Table 1: Comparison with other recent SoA methods (%)

| Dataset | 300W-test-all | | | Menpo-frontal | | | Menpo-profile | | | COFW-68 test | | |
|---|---|---|---|---|---|---|---|---|---|---|---|---|
| Method / Metric | NME | AUC | FR | NME | AUC | FR | NME | AUC | FR | NME | AUC | FR |
| FAN (reported in [5]) | - | 66.9 | - | - | 67.5 | - | - | - | - | - | - | - |
| SAN [1] | 2.86 | 59.7 | 1.00 | 2.95 | 61.9 | 3.11 | 11.71 | 20.7 | 48.39 | 3.50 | 51.9 | 3.94 |
| Our method | 2.25 | 67.8 | 0.17 | 2.16 | 69.0 | 0.18 | 4.71 | 49.0 | 24.30 | 2.65 | 61.8 | 0.00 |
| Dataset | 300VW-category1 | | | 300VW-category2 | | | 300VW-category3 | | | | | |
| FAN (reported in [5]) | - | 72.1 | - | - | 71.2 | - | - | 64.1 | - | | | |
| SAN [1] | 2.58 | 64.5 | 1.10 | 2.57 | 63.2 | 0.42 | 4.06 | 52.9 | 7.19 | | | |
| Our method | 2.08 | 70.9 | 0.29 | 2.07 | 70.1 | 0.04 | 2.54 | 67.4 | 2.01 | | | |

In Table 2, we compare with some most recent best results reported, in the 300W protocol that trains on LFPW-train, HELEN-train, AFW and tests on LFPW-test, HELEN-test, ibug and use NME normalized with inter-ocular distance as the metric.

Table 2: Comparison with SoA methods on 300W dataset using 300W protocol (NME normalized with inter-ocular distance %)

| Method / Subset | Com. | Chal. | Full |
|---|---|---|---|
| SA [3] | 3.45 | 6.38 | 4.02 |
| Wing [2] | **3.27** | 7.18 | 4.04 |
| SAN [1] | 3.34 | 6.60 | 3.98 |
| Our method | 3.33 | **6.29** | **3.91** |

**1. Reviewer 1 (R1)**

1.1 *Original reported performance score.* We also listed the performance score (AUC) reported in the original paper [5] in Table 1. The discrepancy may be caused by different versions of official code.

1.2 *Original benchmark protocol.* To compare with other SoA methods, we reported performance under the original benchmark protocol (300W protocol) widely used by other works in Table 2.

1.3 *How the 3D model was acquired.* Following CE-CLM [50], the 3D model is inferred from 2D annotations using structure from motion. We also tried to use models learned from 3D datasets (e.g. BP4D and Facewarehouse) but found the annotation scheme discrepancy led to inaccurate results.

**2. Reviewer 2 (R2)**

2.1 *Novelty.* Although there are works on CNN-CRF, especially for image segmentation and body landmark detection, there are very few works applying CNN-CRF to facial landmark detection. Compared to body landmarks, facial landmarks have many more points and require accurate localization on the facial contour, thus existing CNN+CRN methods on body landmarks are impractical or not accurate enough to be directly applied to facial landmark detection. Theoretically, our model differs from existing CNN-CRF methods in explicitly employing a fully connected (rather a tree) CRF model and a pose-dependent instead of a fixed pairwise energy function to capture structural relationship variations caused by head pose and deformation. And we perform exact conditional learning and inference compared to widely used approximate methods like mean-field and therefore we have more accurate estimation of the full covariance matrix, which quantifies the structured aleatoric uncertainty. Both R1 and R3 acknowledge the novelties of our model.

2.2 *Citation and comparison with SoA.* We provide comparison with SoA methods in Table 1 and Table 2. For other related work, we will discuss and compare them in the revised paper.

**3. Reviewer 3 (R3)**

3.1 *Unclear experimental methodology.* The baseline is FAN [5] since it performs best among the compared methods. And it uses the same methodology that pretrains the model using 300W-LP. This is applied to all experiments in our original paper where we train one model and test it on all datasets. For a fair comparison, we conduct additional experiment to compare with most recent SoA methods following the same 300W protocol and metrics in Table 2.

3.2 *Missing link to similar work on Continuous CRF.* Different from the two existing continuous CRF works listed by the reviewer, our work models the unary potential weight as dependent on the input, which captures structured heteroscedastic aleatoric uncertainty. Besides, compared to Continuous Conditional Neural Fields (CCNF) applied to face landmark detection, our structured output is directly defined as the location of the facial landmark while in the CCNF it is the probability of the certain landmark being aligned at each pixel location of the image. We will cite and discuss the continuous CRF papers the reviewer mentioned in the revised paper.

3.3 *Clarification on Gaussian NLL.* Gaussian NLL refers to Gaussian negative log likelihood, computed from the mean and covariance from the softmax probability map and embedded in the unary potential. It is the loss when $C_{ij} = \mathbf{0}$.

# References

[1] X. Dong et al. Style aggregated network for facial landmark detection. In *CVPR*, 2018.

[2] Z. Feng et al. Wing loss for robust facial landmark localisation with convolutional neural networks. In *CVPR*, 2018.

[3] Z. Liu et al. Semantic alignment: Finding semantically consistent ground-truth for facial landmark detection. In *CVPR*, 2019.


[Meta-Review · NeurIPS 2019]

The authors did a good job of addressing reviewer concerns in their rebuttal and the reviewers converged towards an accept recommendation.